# Clinical Characteristics and Associated Risk Factors of Prediabetes in the Southwestern Region of Korea from 2010–2019

**DOI:** 10.3390/jcm9041114

**Published:** 2020-04-13

**Authors:** Mi-Ra Oh, Su-Jin Jung, Eun-Ju Bae, Byung-Hyun Park, Soo-Wan Chae

**Affiliations:** 1Clinical Trial Center for Functional Foods, Chonbuk National University Hospital, Jeonju, Jeonbuk 54907, Korea; mroh@jbctc.org (M.-R.O.); sjjeong@jbctc.org (S.-J.J.); 2College of Pharmacy, Chonbuk National University, Jeonju, Jeonbuk 54896, Korea; ejbae7@jbnu.ac.kr; 3Department of Biochemistry and Molecular Biology, Chonbuk National University Medical School, Jeonju, Jeonbuk 54896, Korea

**Keywords:** prediabetes, risk factor, isolated impaired fasting glucose (I-IFG), isolated impaired glucose tolerance (I-IGT), combined I-IFG and I-IGT

## Abstract

This study investigated the clinical characteristics and associated risk factors of prediabetes in the southwestern region of Korea. A total of 323 subjects from 13 prediabetes studies were included in the data analysis. Subjects with prediabetes were divided into the following subtypes: (1) normal glucose tolerance (NGT) with HbA1c 5.7%–6.4%; (2) isolated impaired fasting glucose (I-IFG); (3) isolated impaired glucose tolerance (I-IGT); and (4) combined I-IFG and I-IGT (C-IFG/IGT). Clinical and biochemical variables were compared among subtypes, and multivariate logistic regression analysis was used to identify risk factors for prediabetes subtypes. The overall proportion of subjects with NGT, I-IFG, I-IGT and C-IFG/IGT was 8.4%, 20.7%, 33.1% and 37.8%, respectively. In men, C-IFG/IGT was the most common subtype, while in women, I-IGT was the most common. The parameters related to dysglycemia, atherosclerosis and liver dysfunction were higher in subjects in the C-IFG/IGT subtype than in other subtypes. Multiple linear regression analysis revealed independent risk factors for increased FPG, 2h-PPG and HbA1c levels. This study identified the clinical features and independent risk factors for prediabetes subtypes.

## 1. Introduction

Prediabetes is an intermediate state in which blood glucose levels are higher than normal but not yet high enough to lead to a diagnosis of type 2 diabetes. Prediabetes has no specific signs or symptoms, but individuals with prediabetes have a higher risk of developing diabetes and microvascular complications [1]. The American Diabetes Association defined prediabetes based on any of three distinct entities: impaired fasting glucose (IFG), defined as fasting plasma glucose (FPG) levels of 100–125 mg/dL; impaired glucose tolerance (IGT), defined as 2 h plasma glucose levels of 140–199 mg/dL after an oral glucose tolerance test (OGTT); or glycosylated hemoglobin (HbA1c) level of 5.7%–6.4% [2]. FPG is usually related to basal insulin secretion by the pancreatic β cells and glucose production in the liver, while glucose levels during the OGTT depend on postprandial insulin secretion and subsequent glucose uptake in target cells which are primarily skeletal muscle cells [3]. Therefore, the diagnostic value of IFG and IGT reflect different pathophysiologies of glucose metabolism, thus implicating different risk factors for developing type 2 diabetes.

However, most epidemiologic surveys and population screenings for DM or prediabetes are based on FPG. In Korea, FPG and HbA1c are recommended as a screening guideline for prediabetes, according to the Korean Centers for Disease Control and Prevention [4]. Although FPG is relatively simple and inexpensive compared with OGTT, IGT remains undetected in many subjects when only FPG is measured, which may eventually lead to underestimates in the prevalence of prediabetes [5]. In addition, the clinical characteristics of prediabetes cannot be determined by FPG alone.

Prediabetes can be prevented, and the risk of diabetes can be decreased if lifestyle intervention or pharmacological treatment is implemented before the development of diabetes [6,7,8,9]. Conversely, undiagnosed, and thus untreated, prediabetes causes substantial public health problems. Nevertheless, little attention has been paid to early detection, which is critical to reduce the long-term healthcare burden. Moreover, the prevalence of prediabetes varies in Korea, China and Japan, countries which share a similar lifestyle [10,11,12], suggesting the existence of ethnic variations for the epidemiological and pathophysiological characteristics of prediabetes. In this study, we compared the prevalence of isolated IFG (I-IFG), isolated IGT (I-IGT) and combined IFG and IGT (C-IFG/IGT) in subjects with prediabetes identified according to plasma glucose levels after an OGTT and HbA1c levels. We also examined the anthropometric and biochemical parameters of the subjects, in order to explore whether the parameters are associated with FPG, 2 h post-challenge plasma glucose (2h-PPG) during OGTT and hemoglobin A1c (HbA1c).

## 2. Subjects and Methods

### 2.1. Data Collection

We retrospectively reviewed the medical records of participants who had undergone an OGTT for evaluation of prediabetes between 2010 and 2019, at the Clinical Trial Center in Chonbuk National University Hospital. A total of 13 studies for prediabetes were conducted at the Clinical Trial Center in Chonbuk National University Hospital. The selection of the participants for this analysis is shown in Figure 1. All study examinations were performed according to standard operating procedures by specifically trained and certified medical technical assistants. Data selection criteria were the following: (1) prediabetes according to the OGTT and (2) absence of DM defined as HbA1c ≥ 6.5%, FPG ≥ 126 mg/dL or 2h-PPG ≥ 200 mg/dL. Exclusion criteria were the following: (1) normal glucose tolerance or HbA1c < 5.7% and (2) OGTT without HbA1c assay.

The subjects in the 13 studies had the following features in common: (1) undiagnosed DM or prediabetes, (2) no history of cardiovascular diseases, (3) not taking glucose-lowering agent or insulin therapy, (4) no lipid metabolism disorders, (5) no acute or chronic inflammatory disease, (6) no abnormal hepatic liver function or renal disease (acute/chronic renal failure or nephrotic syndrome) and (7) not currently pregnant or breastfeeding. All subjects gave written informed consent for laboratory analyses, clinical examinations and the use of data records for research purposes.

The information obtained on the demographic and clinical variables were the following: age, sex, height, weight, waist and hip circumference, current smoker, current drinker, alcohol consumption, systolic blood pressure (SBP), diastolic blood pressure (DBP) and pulse. Body mass index (BMI) was defined as weight (kg) divided by height (m^2^). Data from the subjects’ biochemical variables that were collected included plasma glucose (0, 30, 60, 90 and 120 min) levels, HbA1c, fasting insulin (FI), C-peptide, total cholesterol, triglyceride (TG), high-density lipoprotein cholesterol (HDL-C), low-density lipoprotein cholesterol (LDL-C), apolipoprotein A1 (ApoA1), apolipoprotein B (ApoB), high-sensitivity C-reactive protein (hs-CRP), gamma glutamyl transpeptidase (γ-GT), aspartate transaminase (AST), alanine transaminase (ALT) and total bilirubin. The study protocol was approved by the Institutional Review Board of Chonbuk National University (2019-12-003).

### 2.2. Definitions

Prediabetes was defined by the ADA criteria on the basis of the glucose level in the 75 g OGTT or HbA1c value [2]. Subjects with prediabetes were divided into four subtypes: (1) NGT with HbA1c 5.7%–6.4%; (2) I-IFG; (3) I-IGT; and (4) C-IFG/IGT. Individuals with a BMI less than 23.0 were classified as not overweight/obese. Those with a BMI greater than or equal to 23.0 but less than 25.0 were classified as overweight, and those with a BMI greater than or equal to 25.0 were classified as obese. Waist and hip circumference were recorded to the nearest cm, and waist–hip ratio (WHR) was defined as a ratio of waist circumference (WC) to hip circumference (HC). Central obesity was defined as WHR ≥ 0.90 for men and 0.85 for women. Hypertension was defined as having an SBP of ≥ 130 mmHg or DBP of ≥ 85 mmHg. Subjects who smoked at least 1 cigarette daily for 1 year or more were defined as current smokers.

### 2.3. Assays and Calculations

A 75 g OGTT was measured by using a standardized protocol. In brief, after an overnight fast, subjects consumed a 75 g glucose solution, and the plasma glucose level was measured at 0, 30, 60, 90 and 120 min. Glucose parameters were evaluated with incremental area under the curve (iAUC), plasma glucose absolute maximum concentrations (C_max_) and time of maximum concentrations (T_max_). The iAUC (which ignored values below the fasting value) was calculated geometrically, using the trapezoidal rule for plasma glucose for each subject [13]. Plasma glucose levels were measured by using the glucose oxidase method, and HbA1c levels were detected by using an automated glycosylated hemoglobin analyzer (ADAMS^TM^ A1c HA-8180, ARKRAY Factory, Shiga, Japan). Insulin resistance (IR) and pancreatic β-cell function were estimated by a homeostasis model assessment, using the following formula [14]: HOMA-IR = [fasting glucose (mg/dL) × fasting insulin (μU/mL)/405] and HOMA-β (%) = [360 × fasting insulin (μU/mL)]/[fasting glucose (mg/dL) − 63]. Insulin sensitivity was estimated by a quantitative insulin sensitivity check index (QUICKI), using the following formula [15]: QUICKI = 1/[log(fasting insulin (μU/mL)) + log(fasting glucose (mg/dL))]. Insulin and C-peptide concentration were measured using a Cobas e601 module (Hitachi High-Technologies, Tokyo, Japan). hs-CRP, γ-GT, AST, ALT and total bilirubin concentrations were measured by using an ADVIA^®^ 2400 chemistry system (Siemens, Bayern, Germany). Lipid parameters (total cholesterol, HDL-C, LDL-C, triglyceride, ApoA1 and ApoB) were measured by using a Hitachi 7600-110^®^ analyzer (Hitachi High-Technologies, Tokyo, Japan). All of the biochemical variable analyses were performed in a centralized laboratory setup.

### 2.4. Statistical Analysis

Data were analyzed by using the statistical software SAS version 9.4 (SAS Institute Inc., Cary, NC, USA). One-way analysis of variance (ANOVA) was performed to compare the clinical and biochemical variables of subjects from the four subtypes. Post hoc analysis was performed, using the Bonferroni method. The Chi-square test was used for the comparison of categorical variables. Data were expressed as mean ± standard deviation (SD) or number (percent). Natural logarithmic transformation was applied to variables with no normal distribution. Respective relationships of FPG, 2h-PPG or HbA1c concentration with various markers were subjected to univariate regression. Independent variables significantly associated with FPG, 2h-PPG or HbA1c concentration in univariate analysis (*p* < 0.05) and potentially confounding parameters were included as independent covariables in multivariate analysis by multiple regression analysis. The relationship of relevant factor to prediabetes subtypes was assessed by binary logistic regression analysis. Odds ratios were estimated, and their confidence interval (CI) was considered to be 95%. Statistical significance was set at *p* < 0.05.

## 3. Results

### 3.1. General Characteristics of the Subjects According to Prediabetes Subtypes

Table 1 shows the general characteristics of the subjects defined by prediabetes classification. The overall proportion of NGT, I-IFG, I-IGT and C-IFG/IGT was 8.4% (*n* = 27), 20.7% (*n* = 67), 33.1% (*n* = 107) and 37.8% (*n* = 122), respectively. The proportion of C-IFG/IGT was higher in men, while I-IGT was more common in women. The results showed that 44% of men and 31.7% of women suffered from C-IFG/IGT, and that 50.9% of men and 56.7% of women suffered from either I-IGT or I-IFG. The age of subjects with prediabetes ranged from 22 to 70 years, with an average age of 51.7 ± 9.3 years, and 45.2% of the prediabetes subjects were aged 50–59 years. The proportions of subjects with C-IFG/IGT was higher than other subtypes in all age groups, except in subjects aged above 60 years. The proportion of central obesity was significantly higher in the I-IGT subtypes compared with other subtypes. No significant differences were detected in age, BMI, WHR, alcohol consumption, SBP and pulse among the subtypes. There were also no differences in hyperglycemia-associated symptoms, e.g., polyuria, polyphagia, polydipsia and recent weight loss, among subtypes. However, the number of current drinkers and values of DBP were significantly higher in subjects in the C-IFG/IGT subtype compared with subjects in either the I-IFG or I-IGT subtypes.

### 3.2. Glucose Parameters in the Subjects Defined by Prediabetes Subtypes

Table 2 shows the results of the OGTT and glucose-related parameters (iAUC_0–2h_, C_max_, T_max_, FI, HOMA-IR, HOMA-β, QUICKI, C-peptide and HbA1c) by prediabetes subtypes. The glycemic status showed the typical pattern of prediabetes subtypes because each subtype was built according to preestablished values. For example, FPG levels in subjects with the I-IFG and C-IFG/IGT subtypes were significantly higher than FPG levels in subjects with the NGT and IGT subtypes, whereas 2h-PPG levels in subjects with the I-IGT and C-IFG/IGT subtypes were significantly higher than 2h-PPG levels in subjects with the NGT and I-IFG subtypes. This pattern was also obvious for iAUC_0–2h_. The peak glucose levels were observed at 1 h after OGTT in all subtypes. The maximum glucose levels (C_max_), FI and HOMA-IR were significantly higher in subjects with the C-IFG/IGT subtype, whereas HOMA-β and QUICKI were significantly lower in subjects with the C-IFG/IGT subtype compared with other subtypes.

The simple linear regression analysis showed that HbA1c levels were positively correlated with either FPG or 2h-PPG levels (Figure 2A,B), whereas FPG was not correlated with 2h-PPG levels (Figure 2C).

### 3.3. Biochemical Characteristics of the Subjects Defined by Prediabetes Subtypes

The biochemical characteristics of the subjects are listed in Table 3. Overall, subjects in the I-IFG and I-IGT subtypes showed an intermediate health profile between subjects in the NGT and C-IFG/IGT subtypes. In particular, subjects in the C-IFG/IGT subtype showed an increased risk of cardiovascular disease, as evidenced by increased LDL/HDL and cholesterol/HDL ratios and decreased HDL and ApoA1 levels. When the analysis was repeated after adjusting for age and sex, the overall results were the same. Subjects in the C-IFG/IGT subtype also exhibited higher plasma enzyme levels relating to the function of the liver and biliary tract (γ-GT, ALT and albumin).

### 3.4. Risk Factors of FPG, 2h-PPG and HbA1c Levels in Subjects with Prediabetes

Table 4, Table 5 and Table 6 indicate the association of FPG, 2h-PPG and HbA1c levels with various clinical parameters. FPG correlated positively with BMI, WHR, glucose-related parameters (C_max_, FI, HOMA-IR, HbA1c and C-peptide) and liver-related parameters (γ-GT, AST and ALT) (Table 4). However, FPG correlated negatively with HOMA-β and QUICKI. Further multiple regression analysis showed that WHR, C_max_, HOMA-IR and HbA1c were independently related factors to FPG levels. The multiple regression model remained similar after adjustment for age and sex.

Moreover, 2h-PPG correlated positively with SBP, DBP and glucose-related parameters (iAUC_0–2h_, C_max_, T_max_ and HbA1c) and negatively with HDL-C (Table 5). The results of the multiple regression analysis showed that glucose-related parameters (iAUC_0–2h_, C_max_ and T_max_) were independently related factors to 2h-PPG levels. WHR showed a positive correlation trend, with no statistical significance (*p* = 0.059). However, when age and sex were considered in the multiple regression model, WHR was an independently related factor to 2h-PPG levels. The multiple regression model remained similar after further adjustment for age and sex.

HbA1c correlated positively with age, glucose-related parameters (FPG, 2h-PPG, iAUC_0–2h_, C_max_ and T_max_), total cholesterol, LDL-C and hs-CRP (Table 6). HbA1c was negatively correlated with alcohol consumption, SBP, DBP, γ-GT and total bilirubin. The results of the multiple regression analysis showed that age, SBP, FPG, C_max_, T_max_, LDL-C and total bilirubin were independently related factors to HbA1c levels. The multiple regression model remained similar after further adjustment for age and sex.

### 3.5. Risk Factors Associated with Prediabetes Subtypes

The risk factors for prediabetes subtypes were considered for sex, age, BMI, central obesity, smoking, drinking, hypertension, lipoprotein ratios (CHOL/HDL, LDL/HDL, TG/HDL and ApoB/A1) and hs-CRP. In the logistic regression analyses, central obesity (OR, 2.99; 95% CI, 1.13 to 7.91) and smoking (OR, 4.58; 95% CI, 1.41 to 14.87) were independent risk factors for I-IFG. For I-IGT, female sex (OR, 2.23; 95% CI, 1.13 to 4.38) was an independent risk factor. Lastly, male sex (OR, 2.03; 95% CI, 1.14 to 3.61), hypertension (OR, 2.71; 95% CI, 1.31 to 5.58) and hs-CRP (OR, 1.21; 95% CI, 1.02 to 1.49) were independent risk factors for C-IFG/IGT (Figure 3).

## 4. Discussion

In this study, the most common subtype of prediabetes in the southwestern region of Korea was C-IFG/IGT (44.0%) for men and I-IGT (42.1%) for women. I-IGT and C-IFG/IGT accounted for 70.9% of all prediabetes subjects, suggesting that screening with a fasting glucose level, without an OGTT, may underestimate the incidence of prediabetes. The higher distribution of subjects with C-IFG/IGT in this study is not similar to the distributions reported by studies performed in other Asian countries [12,16]. For example, I-IFG subtype is higher than other subtypes in Japan [17], whereas, in China, I-IGT subtype is the highest [18]. It is not clear whether this discrepancy reflects ethnic differences between studies, or whether this result was due to a small sample size. Given that the incidence of IGT is closely related to an unhealthy diet and physical inactivity [19], the higher percentage of subjects with C-IFG/IGT and I-IGT seems to be related to the Westernized lifestyle in Korea. Indeed, 78% of the study subjects were classified as either overweight or obese (as measured by BMI), and 68% of the subjects had central obesity (as measured by WHR). The progression rate from C-IFG/IGT to diabetes was reported to be 2–3 times higher than that from I-IFG or I-IGT to diabetes [20,21,22]. Therefore, relatively higher proportions of the isolated and combined forms of IGT in prediabetes subjects may increase their future risk of type 2 diabetes, which is a well-established risk for life-threatening cardio-metabolic diseases. Therefore, prompt intervention is required to delay or prevent its progression to diabetes, especially for subjects with C-IFG/IGT.

There were sex and age differences in the proportions of subjects with prediabetes in the present study. I-IFG was higher than I-IGT in men, while I-IGT was more common in women. This is not surprising, because the fasting glucose level is positively correlated with alcohol consumption, and post-load glucose levels are predominantly affected by muscle mass. To support this notion, Roh et al. [23] reported that higher-than-average alcohol consumption was associated with a higher odd of IFG in Korean men compared with those with lower alcohol consumption. Smaller skeletal muscle mass with larger visceral fat mass in women seems to be related to the higher glucose levels after glucose loading. The rate of prediabetes increased with age from 10.5% in those aged 20–39 years to 27.3% in those aged 40–49 years. Those aged 50–59 years had the highest rate of prediabetes at 41.5%, and prediabetes decreased in subjects aged above 60 years. Interestingly, the proportions of subjects with C-IFG/IGT was higher than the other subtypes in all age groups, except age above 60 years. Therefore, rates of prediabetes in easily identifiable groups, such as patients in their 50 s, are noticeably higher and suggest that targeted screening and an appropriate healthcare policy might be desirable.

In this study, we investigated the relationship among FPG, 2h-PPG and HbA1c as a function of glucose intolerance in subjects. We found a strong correlation between HbA1c and either FPG or 2h-PPG, but only a small correlation between FPG and 2h-PPG. These results are in accordance with previous studies that demonstrated a moderate-to-strong correlation between HbA1c and FPG, as well as 2h-PPG [24,25]. However, in contrast to this study, Van’t Riet and colleagues reported that only 39% of subjects with prediabetes and HbA1c values in the range of 5.7%–6.4% were diagnosed with prediabetes based on the results of a 75 g OGTT [26]. Possible mechanisms for this discrepancy may be differences in race, genetic trait, sex composition, age and socioeconomic status between subjects in different studies. Considering that HbA1c simply represents an average blood sugar level over the previous three months, it is clear that, when tested alone, HbA1c can misclassify some subjects to either the normal or the diabetes group. The weak correlation of FPG with 2h-PPG observed in this study may imply that fasting and post-load glucose levels reflect different metabolic processes, especially in the prediabetic range of glucose tolerance. More specifically, FPG reflects hepatic insulin resistance with basal insulin secretion, whereas the 2h-PPG level predominantly depends on muscular glucose uptake and postprandial insulin secretion [3].

Based on multiple regression analysis after adjusting age and sex, we confirmed that the WHR, a surrogate marker for central obesity, is an independent risk factor for FPG and 2h-PPG levels, but not for the HbA1c level. Although the subjects in the present study had a much lower BMI than Western people, Koreans have a similar rate of diabetes prevalence as residents of Western countries. One possible explanation is that Korean people have less muscle and more visceral fat than Western people. Visceral fat is a main source of inflammatory cytokines and metabolites, which result in hepatic and systemic insulin resistance. Therefore, WHR is a useful marker to predict I-IFG and I-IGT. Additionally, smoking was associated with the odds of having I-IFG. Subjects who were current smokers were 4–5 times more likely to have I-IFG compared with subjects in other groups. These results may be due to the higher proportion of men in the I-IFG groups. Similar to these results, longitudinal studies have reported that cigarette smoking is an independent risk factor for type 2 diabetes [27,28].

Another interesting finding of the present study is that subjects in the NGT subtype had normal levels of biochemical and clinical parameters. The results indicated that about 8.4% of study subjects who had elevated HbA1c values exhibited normal glucose levels during OGTT. In 2010, the American Diabetes Association (ADA) proposed the use of HbA1c for the diagnosis of prediabetes and recommended cut-off values of between 5.7% and 6.4% [2]. However, several studies published since the 2010 ADA statement have questioned the value of HbA1c for the diagnosis of prediabetes [24,25,26]. HbA1c has several advantages for the assessment of glucose metabolism that may favor its use for diagnostic purposes, but there are patient-dependent and independent factors, some of which are not well defined [29]. Therefore, HbA1c alone without OGTT may miss the diagnoses of prediabetes and diabetes.

There are several limitations in this study. First, it may have a potential selection bias. Selection bias occurs when recruiters selectively enroll subjects into a trial based on protocol. However, the data analyzed in this study are based on simple randomized clinical trials. Given that simple randomization is the simplest and most effective method to prevent selection bias [30], we believe that, even if there is a selection bias, it may be minimal. Second, the sample size used in this study was relatively small compared to other studies conducted elsewhere, and the smaller size may cause biased results. Therefore, caution should be exercised in interpreting the results as an indication of the presence of prediabetes. Despite these limitations, this is the first study to identify characteristics and associated risk factors of prediabetes subtypes in Korean subjects with prediabetes. We believe that the findings described in this study extend the present knowledge of the epidemiological and pathophysiological characteristics of prediabetes subtypes in the Korean population.

## Figures and Tables

**Figure 1 jcm-09-01114-f001:**
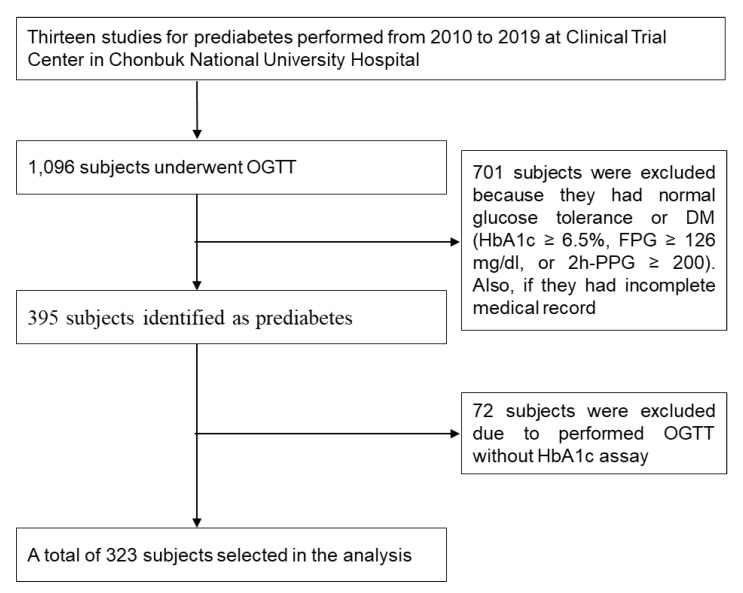
Flow diagram of data collection.

**Figure 2 jcm-09-01114-f002:**
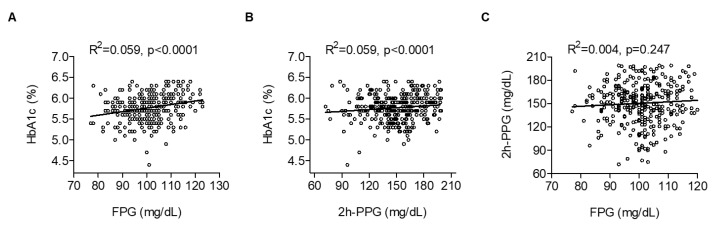
Correlation among (**A**) FPG, (**B**) 2h-PPG and HbA1c, and (**C**) FPG and 2h-PPG.

**Figure 3 jcm-09-01114-f003:**
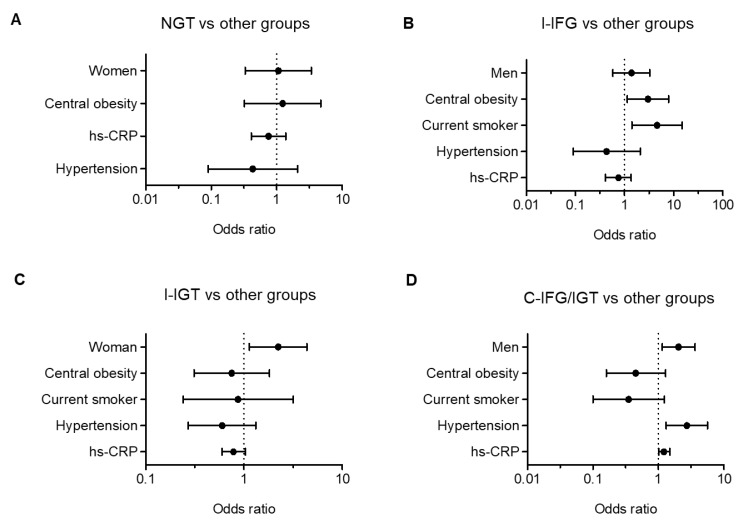
Logistic regression between (**A**) NGT vs .other prediabetes groups, (**B**) I-IFG vs. other prediabetes groups, (**C**) I-IGT vs. other prediabetes groups, and (**D**) C-IFG/IGT vs. other prediabetes groups.

**Table 1 jcm-09-01114-t001:** General characteristics of the subjects according to prediabetes subtypes.

	NGT	I-IFG	I-IGT	C-IFG/IGT	Total	*p*-Value ^†^
Total	27 (8.4)	67 (20.7)	107 (33.1)	122 (37.8)	323 (100)	<0.0001
Men, *n* (%)	8 (29.6)	43 (64.2)	38 (35.5)	70 (57.4)	159 (49.2)	<0.0001
Women, *n* (%)	19 (70.4)	24 (35.8)	69 (64.5)	52 (42.6)	164 (50.8)	<0.0001
Age (years)	51.4 ± 8.6	51.0 ± 9.3	53.0±9.4	51.1 ± 9.3	51.7 ± 9.3	0.410
≤39 years, *n* (%)	3 (11.1)	8 (11.9)	10 (9.3)	13 (10.7)	34 (10.5)	0.937
40–49 years, *n* (%)	7 (25.9)	18 (26.9)	22 (20.6)	29 (23.8)	76 (23.5)	
50–59 years, *n* (%)	12 (44.4)	29 (43.3)	47 (43.9)	58 (47.5)	146 (45.2)	
≥60 years, *n* (%)	5 (18.5)	12 (17.9)	28 (26.2)	22 (18.0)	67 (20.7)	
BMI (kg/m^2^)	24.3 ± 2.6	25.3 ± 2.6	24.7 ± 2.4	25.3 ± 2.4	25.0 ± 2.5	0.084
Normal weight, *n* (%)	10 (37.0)	15 (22.4)	26 (24.3)	20 (16.4)	71 (22.0)	0.364
Overweight, *n* (%)	7 (25.9)	18 (26.9)	32 (29.9)	40 (32.8)	97 (30.0)	
Obese, *n* (%)	10 (37.0)	34 (50.7)	49 (45.8)	62 (50.8)	155 (48.0)	
Waist–hip ratio						
Men	0.93 ± 0.03	0.92 ± 0.05	0.92 ± 0.05	0.94 ± 0.05	0.93 ± 0.05	0.226
Women	0.88 ± 0.07	0.91 ± 0.05	0.91 ± 0.05	0.92 ± 0.04	0.91 ± 0.05	0.153
Central obesity, *n* (%)	16 (59.3)	36 (53.7)	80 (74.8)	88 (72.1)	220 (68.1)	0.049
Current smoker, *n* (%)	0 (0.0)	7 (10.4)	6 (5.6)	7 (5.7)	20 (6.2)	0.085
Current drinker, *n* (%)	5 (18.5)	25 (37.3)	40 (37.4)	62 (50.8)	132 (40.9)	0.001
Alcohol consumption (unit/weeks)	4.66 ± 1.37	8.1 ± 7.4	6.3 ± 6.9	7.3 ± 6.8	7.1 ± 6.8	0.623
SBP (mmHg)	119.2 ± 12.5	123.1 ± 12.5	123.0 ± 13.4	125.1 ± 12.9	123.5 ± 13.0	0.163
DBP (mmHg)	76.74 ± 10.23	77.9 ± 9.2	78.0 ± 9.5	80.8 ± 9.2	78.9 ± 9.4	0.043 ^a^
Pulse (BPM)	72.78 ± 7.56	70.8 ± 9.9	70.3 ± 9.1	70.4 ± 8.1	70.7 ± 8.8	0.604

BMI, body mass index; SBP, systolic blood pressure; DBP, diastolic blood pressure. Continuous variables are presented as mean ± SD. Categorical variables are presented as number and percentage. ^†^ Categorical variables were compared using Chi-square test. Continuous variables were compared using one-way ANOVA. The post hoc Bonferroni was used for comparison among groups and showed significant differences for the following: ^a^
*p* < 0.05: NGT vs. C-IFG/IGT.

**Table 2 jcm-09-01114-t002:** Various parameters of dysglycemia among prediabetes subtypes.

	NGT	I-IFG	I-IGT	C-IFG/IGT	*p*-Value ^†^	*p*-Value ^‡^
OGTT						
FPG (mg/dL)	92.1 ± 5.1	105.4 ± 5.0	92.8 ± 5.1	107.8 ± 6.1	<0.0001	<0.0001 ^a^
0.5h-PPG (mg/dL)	152.1 ± 28.7	176.1 ± 27.6	163.7 ± 21.8	183.0 ± 23.5	<0.0001	<0.0001 ^b^
1h-PPG (mg/dL)	157.2 ± 35.3	181.3 ± 37.8	181.0 ± 27.6	201.5 ± 35.3	<0.0001	<0.0001 ^c^
1.5h-PPG (mg/dL)	139.2 ± 28.3	151.6 ± 28.3	173.5 ± 28.8	189.6 ± 33.6	<0.0001	<0.0001 ^d^
2h-PPG (mg/dL)	118.0 ± 16.4	118.3 ± 16.7	162.1 ± 15.5	165.3 ± 17.3	<0.0001	<0.0001 ^e^
iAUC_0–2h_ (h*mg/dL)	93.6 ± 36.1	100.7 ± 37.1	137.3 ± 33.0	140.5 ± 39.3	<0.0001	<0.0001 ^f^
C_max_ (mg/dL)	171.0 ± 23.5	190.9 ± 30.6	189.9 ± 21.2	209.9 ± 28.9	<0.0001	<0.0001 ^g^
T_max_ (min)	60 [30–90]	60 [30–90]	60 [30–120]	60 [30–120]	<0.0001	<0.0001 ^h^
FI (μU/mL)	6.4 ± 2.6	8.0 ± 5.6	7.2 ± 4.3	8.7 ± 5.2	0.041	0.019 ^i^
HOMA-IR	1.5 ± 0.6	2.1 ± 1.5	1.7 ± 1.0	2.3 ± 1.4	<0.0001	<0.0001 ^j^
HOMA-β (%)	79.0 ± 29.0	68.6 ± 47.5	89.6 ± 53.5	71.3 ± 42.7	0.009	0.025 ^k^
QUICKI ^§^	0.37 ± 0.03	0.36 ± 0.06	0.37 ± 0.06	0.35 ± 0.03	0.001	0.001 ^l^
C-peptide (ng/mL)	1.8 ± 0.4	2.1 ± 0.9	1.9 ± 0.6	2.1 ± 0.7	0.070	0.055
HbA1c (%)	5.9 ± 0.2	5.8 ± 0.4	5.7 ± 0.3	5.8 ± 0.3	<0.0001	<0.0001 ^m^

FPG, fasting plasma glucose; PPG, postprandial plasma glucose; iAUC, incremental area under the curve; C_max_, maximum concentration; T_max_, time to C_max_; FI, fasting insulin; HOMA-IR, homeostatic model assessment for insulin resistance; HOMA-β, homeostatic model assessment for β-cell function; QUICKI, quantitative insulin sensitivity check index. T_max_ is presented as median mean (range). Data are presented as mean ± SD. ^†^ Comparisons were performed by one-way ANOVA. ^‡^ The *p*-value was adjusted for age and sex. ^§^ Log-transformed data were used for analysis. The post hoc Bonferroni was used for comparison among groups and showed significant differences for the following: all comparisons are *p* < 0.05; ^a^ NGT vs. I-IFG; NGT vs. C-IFG/IGT; I-IFG vs. I-IGT; I-IGT vs. C-IFG/IGT; ^b^ NGT vs. I-IFG; NGT vs. C-IFG/IGT; I-IFG vs. I-IGT; ^c^ NGT vs. I-IFG; NGT vs. I-IGT; NGT vs. C-IFG/IGT; I-IFG vs. C-IFG/IGT; I-IGT vs. C-IFG/IGT; ^d^ NGT vs. I-IGT; NGT vs. C-IFG/IGT; I-IFG vs. I-IGT; I-IFG vs. C-IFG/IGT; I-IGT vs. C-IFG/IGT; ^e^ NGT vs. I-IGT; NGT vs. C-IFG/IGT; I-IFG vs. I-IGT; I-IFG vs. C-IFG/IGT; ^f^ NGT vs. I-IGT; NGT vs. C-IFG/IGT; I-IFG vs. I-IGT, I-IFG vs. C-IFG/IGT; ^g^ NGT vs. I-IFG; NGT vs. C-IFG/IGT; I-IFG vs. C-IFG/IGT; I-IGT vs. C-IFG/IGT; ^h^ NGT vs. I-IGT; I-IFG vs. I-IGT; I-IFG vs. C-IFG/IGT; ^i^ NGT vs. C-IFG/IGT; ^j^ NGT vs. C-IFG/IGT; I-IGT vs. C-IFG/IGT; ^k^ I-IGT vs. C-IFG/IGT; ^l^ NGT vs. C-IFG/IGT; I-IGT vs. C-IFG/IGT; ^m^ NGT vs. I-IGT; I-IFG vs. I-IGT; I-IGT vs. C-IFG/IGT.

**Table 3 jcm-09-01114-t003:** Biochemical characteristics among prediabetes subtypes.

	NGT	I-IFG	I-IGT	C-IFG/IGT	*p*-Value ^†^	*p*-Value ^‡^
CHOL (mg/dL)	194.2 ± 30.4	191.2 ± 32.3	197.3 ± 34.3	201.2 ± 30.7	0.272	0.220
TG (mg/dL)	116.2 ± 64.1	136.4 ± 78.2	141.3 ± 70.0	149.2 ± 74.4	0.191	0.194
HDL-C (mg/dL)	53.9 ± 10.8	50.4 ± 10.2	48.0 ± 10.6	48.0 ± 10.2	0.028	0.030 ^a^
LDL-C (mg/dL)	117.8 ± 29.7	114.9 ± 28.1	120.8 ± 32.7	125.4 ± 29.7	0.179	0.158
CHOL/HDL	3.75 ± 0.96	3.91 ± 0.85	4.29 ± 1.10	4.34 ± 0.97	0.004	0.005 ^b^
LDL/HDL	2.28 ± 0.76	2.37 ± 0.72	2.65 ± 0.93	2.69 ± 0.76	0.019	0.020 ^c^
TG/HDL	2.39 ± 1.91	2.83 ± 1.69	3.25 ± 1.99	3.39 ± 1.95	0.053	0.053
ApoA1 (g/L)	1.52 ± 0.23	1.47 ± 0.20	1.38 ± 0.20	1.38 ± 0.20	0.025	0.024 ^d^
ApoB (g/L)	1.12 ± 0.16	1.09 ± 0.23	1.06 ± 0.25	1.09 ± 0.22	0.813	0.735
ApoB/ApoA1	0.75 ± 0.14	0.75 ± 0.18	0.80 ± 0.21	0.81 ± 0.20	0.451	0.360
hs-CRP ^§^ (mg/L)	0.53 ± 0.70	1.01 ± 1.91	0.75 ± 1.23	1.32 ± 2.50	0.434	0.415
γ-GT (IU/L)	21.0 ± 10.8	38.6 ± 30.2	26.5 ± 19.1	35.6 ± 26.0	0.001	0.156
AST (IU/L)	21.4 ± 4.5	24.2 ± 7.0	23.4 ± 6.3	24.2 ± 6.7	0.204	0.413
ALT (IU/L)	20.8 ± 7.5	24.9 ± 11.0	22.6 ± 8.7	26.0 ± 12.1	0.025	0.129
Total bilirubin (mg/dL)	0.75 ± 0.26	0.84 ± 0.32	0.85 ± 0.37	0.94 ± 0.43	0.201	0.257

CHOL, cholesterol; TG, triglyceride; hs-CRP, high-sensitivity C-reactive protein; ALP, alkaline phosphatase; AST, aspartate transaminase; ALT; γ-GT, gamma glutamyl transpeptidase; CBC, complete blood count; WBC, white blood cells; RBC, red blood cells; BUN, blood urea nitrogen; CK, creatine kinase. Data are presented as mean ± SD. ^†^ Comparisons were performed by one-way ANOVA. ^‡^ The *p*-value was adjusted for age and sex. ^§^ Log-transformed data were used for analysis. The post hoc Bonferroni was used for comparison among groups and showed significant differences for the following: ^a^ NGT vs. C-IFG/IGT; ^b^ NGT vs. C-IFG/IGT; I-IFG vs. C-IFG/IGT; ^c^ NGT vs. C-IFG/IGT; ^d^ NGT vs. C-IFG/IGT.

**Table 4 jcm-09-01114-t004:** Factors related to elevating FPG levels in subjects with prediabetes.

Variable	Simple Linear Regression		Multiple Regression			
			Model 1 ^†^		Model 2 ^‡^	
	r	*p*-Value	*Β*	*p*-Value	*β*	*p*-Value
Age	−0.051	0.359	-	-	-	-
BMI	0.155	0.005	−0.043	0.497	−0.048	0.449
WHR	0.216	<0.0001	0.162	0.005	0.169	0.005
Alcohol consumption	0.065	0.462	-	-	-	-
SBP	0.080	0.154	-	-	-	-
DBP	0.103	0.065	-	-	-	-
Pulse	0.003	0.964	-	-	-	-
2h-PPG	0.065	0.247	-	-	-	-
iAUC_0–2h_	0.043	0.454	-	-	-	-
C_max_	0.426	<0.0001	0.348	<0.0001	0.346	<0.0001
T_max_	−0.096	0.095	-	-	-	-
Fasting insulin ^a^	0.165	0.003	-	-	-	-
HOMA-IR	0.286	<0.0001	0.223	<0.0001	0.216	0.001
HOMA-β	−0.233	<0.0001	-	-	-	-
QUICKI ^a,§^	−0.255	<0.0001	-	-	-	-
C-peptide ^a^	0.183	0.007	-	-	-	-
HbA1c	0.242	<0.0001	0.227	<0.0001	0.237	<0.0001
Total CHOL	0.025	0.661	-	-	-	-
TG	0.044	0.451	-	-	-	-
HDL-C	−0.027	0.646	-	-	-	-
LDL-C	0.019	0.741	-	-	-	-
Apo B/Apo A1	−0.019	0.809	-	-	-	-
hs-CRP ^§^	0.080	0.233	-	-	-	-
γ-GT	0.230	<0.0001	0.045	0.436	0.048	0.415
AST ^a^	0.116	0.037	-	-	-	-
ALT ^a^	0.165	0.003	-	-	-	-
Total bilirubin ^a^	0.129	0.074	-	-	-	-

^a^ Did not enter into the multiple regression due to its multicollinearity. ^§^ Log-transformed data were used for analysis. ^†^ Model 1 was not adjusted for any variable. ^‡^ Model 2 was adjusted for age and sex.

**Table 5 jcm-09-01114-t005:** Factors related to elevating 2h-PPG levels in subjects with prediabetes.

Variable	Simple Linear Regression		Multiple Regression			
			Model 1 ^†^		Model 2 ^‡^	
	r	*p*-Value	*β*	*p*-Value	*β*	*p*-Value
Age	0.070	0.211	-	-	-	-
BMI	0.047	0.406	-	-	-	-
WHR	0.116	0.059	0.089	0.072	0.099	0.048
Alcohol consumption	0.014	0.873	-	-		
SBP	0.130	0.019	0.077	0.119	0.087	0.081
DBP ^a^	0.132	0.017	-	-	-	-
Pulse	−0.015	0.794	-	-	-	-
FPG	0.065	0.247	-	-	-	-
iAUC_0–2h_	0.580	<0.0001	0.734	<0.0001	0.705	<0.0001
C_max_	0.376	<0.0001	−0.320	<0.0001	−0.282	0.003
T_max_	0.420	<0.0001	0.258	<0.0001	0.270	<0.0001
Fasting insulin	0.052	0.354	-	-	-	-
HOMA-IR	0.058	0.297	-	-	-	-
HOMA-β	0.041	0.468	-	-	-	-
QUICKI ^§^	−0.073	0.192	-	-	-	-
C-peptide	0.037	0.594	-	-	-	-
HbA1c	0.119	0.032	0.042	0.395	0.023	0.678
Total CHOL	0.052	0.368	-	-	-	-
TG	0.086	0.139	-	-	-	-
HDL-C	−0.120	0.038	−0.050	0.302	−0.049	0.313
LDL-C	0.081	0.162	-	-	-	-
Apo B/Apo A1	0.020	0.794	-	-	-	-
hs-CRP ^§^	0.070	0.295	-	-	-	-
γ-GT	−0.037	0.534	-	-	-	-
AST	0.031	0.577	-	-	-	-
ALT	0.067	0.228	-	-	-	-
Total bilirubin	−0.001	0.994	-	-	-	-

^a^ Did not enter into the multiple regression due to its multicollinearity. ^§^ Log-transformed data were used for analysis. ^†^ Model 1 was not adjusted for any variable. ^‡^ Model 2 was adjusted for age and sex.

**Table 6 jcm-09-01114-t006:** Factors related to elevating HbA1c levels in subjects with prediabetes.

Variable	Simple Linear Regression		Multiple Regression			
			Model 1 ^†^		Model 2 ^‡^	
	r	*p*-Value	*β*	*p*-Value	*β*	*p*-Value
Age	0.223	<0.0001	0.130	0.009	0.127	0.010
BMI	0.064	0.254	-	-	-	-
WHR	0.018	0.771	-	-	-	-
Alcohol consumption	−0.193	0.028	−0.074	0.201	−0.044	0.456
SBP	−0.166	0.003	−0.139	0.006	−0.125	0.014
DBP ^a^	−0.145	0.009	-	-	-	
Pulse	−0.044	0.428	-	-	-	
FPG	0.242	<0.0001	0.287	<0.0001	0.289	<0.0001
2h-PPG	0.119	0.032	0.029	0.606	0.014	0.809
iAUC_0–2h_ ^a^	0.191	0.001	-	-	-	-
C_max_	0.238	<0.0001	0.182	0.002	0.187	0.001
T_max_	0.144	0.013	0.132	0.012	0.140	0.008
Fasting insulin	0.071	0.202	-	-	-	-
HOMA-IR	0.099	0.076	-	-	-	-
HOMA-β	−0.011	0.840	-	-	-	-
QUICKI ^§^	−0.030	0.591	-	-	-	-
C-peptide,	−0.030	0.660	-	-	-	-
Total CHOL ^a^	0.160	0.006	-	-	-	-
TG	−0.025	0.667	-	-	-	-
HDL-C	0.008	0.885	-	-	-	-
LDL-C	0.211	<0.0001	0.143	0.004	0.133	0.007
Apo A1	0.071	0.366				
Apo B	0.145	0.063				
hs-CRP ^§^	0.117	0.036	0.043	0.378	0.055	0.252
γ-GT	−0.194	0.001	−0.048	0.683	−0.041	0.474
AST	0.047	0.403	-	-	-	-
ALT	0.087	0.119	-	-	-	-
Total bilirubin	−0.153	0.033	0.136	0.007	0.136	0.006

^a^ Did not enter into the multiple regression due to its multicollinearity. ^§^ Log-transformed data were used for analysis. ^†^ Model 1 was not adjusted for any variable. ^‡^ Model 2 was adjusted for age and sex.

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
