# Peer review of "Clinical Characteristics and Associated Risk Factors of Prediabetes in the Southwestern Region of Korea from 2010–2019"

_jcm, 2020, doi:10.3390/jcm9041114_

Round 1
Reviewer 1 Report
This is a well-written, nicely performed study. I have just a few minor comments.
<Comments>
Introduction:
The reversal of impaired glucose tolerance is questioned. Looks like we are able to delay the progression of it to diabetes utmost.
Abstract and Results:
This is a purely observational study so that the following statements/wording are not adequate.
- Abstract, the last sentence, "...should be considered to delay..". The authors did not show us the the data to support this statement.
- Titles of Tables 4 and 5, the word "responsible for" is too much. "related to" would be OK.
<References>
Insulin secretion and insulin sensitivity on the oral glucose tolerance test (OGTT) in middle-aged Japanese. Endocr J. 2012;59(1):55-64.
This paper, a similar study in Japanese subjects, is highly relevant to the topic and should be cited.
Author Response
This is a well-written, nicely performed study. I have just a few minor comments.
Introduction:
The reversal of impaired glucose tolerance is questioned. Looks like we are able to delay the progression of it to diabetes utmost.
--> Lifestyle and pharmacological interventions have been shown to delay or even prevent T2DM in those with pre-diabetes (see below references). Thus one approach to tackling the increasing prevalence of T2DM is to identify those with pre-diabetes and offer such intervention. We cited below references and rephrased the sentence as follows:
Prediabetes can be reversed prevented, and the risk of diabetes can be decreased if lifestyle intervention or pharmacological treatment is implemented before the development of diabetes [6-9].
References
- Chiasson J, Josse R, Gomis R, Hanefeld M, Karasik A, Laakso M. Acarbose for prevention of type 2 diabetes mellitus: the STOP-NIDDM randomised trial. Lancet 2002;359(9323):2072– 7.
- Gillies C, Abrams K, Lambert P, Cooper N, Sutton A, Hsu T, et al. Pharmacological and lifestyle interventions to prevent or delay type 2 diabetes in people with impaired glucose tolerance: systematic review and meta-analysis. BMJ 2007;334:299.
Abstract and Results:
This is a purely observational study so that the following statements/wording are not adequate.
- Abstract, the last sentence, "...should be considered to delay..". The authors did not show us the data to support this statement.
--> As commented, we deleted the last sentence in Abstract.
A lifestyle modification program targeted to specific prediabetes subtypes should be considered to delay progression to diabetes.
- Titles of Tables 4 and 5, the word "responsible for" is too much. "related to" would be OK.
--> As commented, we changed titles of Tables 4, 5 and 6.
References
Insulin secretion and insulin sensitivity on the oral glucose tolerance test (OGTT) in middle-aged Japanese. Oka R, Yagi K, Sakurai M, Nakamura K, Moriuchi T, Miyamoto S, Nohara A, Kawashiri MA, Takeda Y, Yamagishi M.Oka R, et al. Endocr J. 2012;59(1):55-64.
This paper, a similar study in Japanese subjects, is highly relevant to the topic and should be cited.
--> As commented, we cited the above reference.
Reviewer 2 Report
Summary
The authors examine the prevalence of IFG, IGT, combined IFG and IGT in patients with prediabetes. They pooled subjects who had undergone an OGTT from 13 studies conducted at the Clinical Trial Center in Chonbuk National University Hospital. The authors found that the most common subtype of prediabetes was C-IFG/IGT for men and I-IGT for women which is different from other Asian countries. They also note a strong correlation between HbA1c and either FPG or 2h-PPG.
General comments:
-the term “prediabetic subjects” should not be used, rather it should be subjects with prediabetes
Specific comments:
Introduction
-Would be helpful to give context of what screening guidelines in Korea recommend in terms of screening patients for prediabetes (e.g., what risk factors, what age, how often, using what test). This would provide context for the discussion where the authors suggest targeted screening in higher risk patients such as those in their 50s.
Discussion
-Authors conclude that the distribution of subtypes of prediabetes differ from other Asian countries but do not explain the distribution in other countries- please add this information and compare/contrast.
-How do the findings compare to studies using more national data such as the Korean NHANES data? Would be helpful to compare the study’s findings to other studies conducted in the Korean population.
-I appreciate the comprehensive analyses that the authors conducted to compare biochemical characteristics of subjects based on prediabetes subtypes. However, I wonder how this information ultimately affects treatment and follow-up of patients. Could this be used to risk stratify patients who would benefit from lifestyle change (such as DPP) vs. medication? Have analyses of the DPP Trial or other larger diabetes prevention trials shown differential benefit of metformin vs. lifestyle change in patients with different prediabetes subtypes?
Author Response
Summary
The authors examine the prevalence of IFG, IGT, combined IFG and IGT in patients with prediabetes. They pooled subjects who had undergone an OGTT from 13 studies conducted at the Clinical Trial Center in Chonbuk National University Hospital. The authors found that the most common subtype of prediabetes was C-IFG/IGT for men and I-IGT for women which is different from other Asian countries. They also note a strong correlation between HbA1c and either FPG or 2h-PPG.
General comments:
-the term “prediabetic subjects” should not be used, rather it should be subjects with prediabetes
--> As commented, we named participants “subjects with prediabetes”.
Specific comments:
Introduction
-Would be helpful to give context of what screening guidelines in Korea recommend in terms of screening patients for prediabetes (e.g., what risk factors, what age, how often, using what test). This would provide context for the discussion where the authors suggest targeted screening in higher risk patients such as those in their 50s.
--> As commented, we described the screening guidelines in Korea as follows:
In Korea, FPG and HbA1c are recommended as a screening guideline for prediabetes according to the Korean Centers for Disease Control and Prevention [4].
Reference
[4]. Evidence-based Guideline for Type 2 Diabetes in Primary Care. Available online: www.digitalcpg.kr (accessed on 7 April 2020).
Discussion
-Authors conclude that the distribution of subtypes of prediabetes differ from other Asian countries but do not explain the distribution in other countries- please add this information and compare/contrast.
--> As commented, we described the subtypes of prediabetes in other Asian countries as follows:
For example, I-IFG subtype is higher than other subtypes in Japan [17], whereas, in China, I-IGT subtype is the highest [18].
References
[17]. Oka, R.; Yagi, K.; Sakurai, M.; Nakamura, K.; Moriuchi, T.; Miyamoto, S.; Nohara, A.; Kawashiri, M.A.; Takeda, Y.; Yamagishi, M. Insulin secretion and insulin sensitivity on the oral glucose tolerance test (OGTT) in middle-aged Japanese. Endocr J 2012, 59, 55-64, doi:10.1507/endocrj.ej11-0157.
[18]. Liu, W.; Hua, L.; Liu, W.F.; Song, H.L.; Dai, X.W.; Yang, J.K. The prevalence of glucose metabolism disturbances in Chinese Muslims and possible risk factors: a study from northwest China. Arq Bras Endocrinol Metabol 2014, 58, 715-723, doi:10.1590/0004-2730000002654.
-How do the findings compare to studies using more national data such as the Korean NHANES data? Would be helpful to compare the study’s findings to other studies conducted in the Korean population.
--> We can get the information of prediabetes prevalence, but not the information of prediabetes subtypes from the Korean NHANES.
-I appreciate the comprehensive analyses that the authors conducted to compare biochemical characteristics of subjects based on prediabetes subtypes. However, I wonder how this information ultimately affects treatment and follow-up of patients. Could this be used to risk stratify patients who would benefit from lifestyle change (such as DPP) vs. medication? Have analyses of the DPP Trial or other larger diabetes prevention trials shown differential benefit of metformin vs. lifestyle change in patients with different prediabetes subtypes?
--> To our knowledge, there are no comparative studies for the differential benefit of lifestyle intervention versus pharmacotherapy among prediabetes subtypes. Landmark studies performed in China, Finland and the United States show that there are abundant evidences of the efficacy of intensive lifestyle programs in reducing the risk of developing type 2 diabetes among those with prediabetes (ref. 1-3). Diabetes Prevention Program (DPP) trial has reported that lifestyle intervention decreases the incidence of type 2 diabetes by 58% compared with 31% in the metformin-treated group (ref. 4). However, one important and inadequately recognized limitation of the lifestyle intervention studies is that most of the trials have focused on subjects with IGT (ref. 5). The small number of trials that have included subjects with impaired fasting glucose (IFG) have shown no benefit for this subgroup unless they also have IGT (ref. 6-8).
References
- Pan XR, Li GW, Hu YH, et al. Effects of diet and exercise in preventing NIDDM in people with impaired glucose tolerance. The Da Qing IGT and diabetes study. Diabetes Care 1997; 20:537–544.
- Tuomilehto J, Lindstrom J, Eriksson JG, et al. Prevention of type 2 diabetes mellitus by changes in lifestyle among subjects with impaired glucose tolerance. N Engl J Med 2001; 344: 1343–1350.
- Knowler WC, Barrett-Connor E, Fowler SE, et al. Reduction in the incidence of type 2 diabetes with lifestyle intervention or metformin. N Engl J Med 2002; 346: 393–403.
- Diabetes Prevention Program Research Group. Long-term effects of lifestyle intervention or metformin on diabetes development and microvascular complications over 15-year follow-up: the diabetes prevention program outcomes study. Lancet Diabetes Endocrinol 2015;3: 866–875.
- Jonathan E. Shaw. Prediabetes: lifestyle, pharmacotherapy or regulation? Ther Adv Endocrinol Metab 2019, Vol. 10: 1–6 DOI: 10.1177/ 2042018819863020
- Thankappan KR, Sathish T, Tapp RJ, et al. A peer-support lifestyle intervention for preventing type 2 diabetes in India: a cluster-randomized controlled trial of the Kerala diabetes prevention program. PLoS Med 2018; 15: e1002575.
- Saito T, Watanabe M, Nishida J, et al. Lifestyle modification and prevention of type 2 diabetes in overweight Japanese with impaired fasting glucose levels: a randomized controlled trial. Arch Intern Med 2011; 171: 1352–1360.
- Weber MB, Ranjani H, Staimez LR, et al. The stepwise approach to diabetes prevention: results from the D-CLIP randomized controlled trial. Diabetes Care 2016; 39: 1760–1767.